# Non-Destructive Testing of Alfalfa Seed Vigor Based on Multispectral Imaging Technology

**DOI:** 10.3390/s22072760

**Published:** 2022-04-03

**Authors:** Shuheng Zhang, Hanguo Zeng, Wei Ji, Kun Yi, Shuangfeng Yang, Peisheng Mao, Zhanjun Wang, Hongqian Yu, Manli Li

**Affiliations:** 1College of Grassland Science and Technology, China Agricultural University, Beijing 100193, China; sy20203243190@cau.edu.cn (S.Z.); xz@cau.edu.cn (H.Z.); s20203243160@cau.edu.cn (W.J.); b20213241039@cau.edu.cn (K.Y.); shuangfengy@cau.edu.cn (S.Y.); maops@cau.edu.cn (P.M.); 2Institute of Desertification Control, Ningxia Academy of Agricultural and Forestry Sciences, Yinchuan 750002, China; nxwzhj@163.com (Z.W.); nxyuhq@163.com (H.Y.)

**Keywords:** seed vigor, multispectral imaging, seed germination, seed viability, multivariate analysis

## Abstract

Seed vigor is an important index to evaluate seed quality in plant species. How to evaluate seed vigor quickly and accurately has always been a serious problem in the seed research field. As a new physical testing method, multispectral technology has many advantages such as high sensitivity and accuracy, nondestructive and rapid application having advantageous prospects in seed quality evaluation. In this study, the morphological and spectral information of 19 wavelengths (365, 405, 430, 450, 470, 490, 515, 540, 570, 590, 630, 645, 660, 690, 780, 850, 880, 940, 970 nm) of alfalfa seeds with different level of maturity and different harvest periods (years), representing different vigor levels and age of seed, were collected by using multispectral imaging. Five multivariate analysis methods including principal component analysis (PCA), linear discriminant analysis (LDA), support vector machine (SVM), random forest (RF) and normalized canonical discriminant analysis (nCDA) were used to distinguish and predict their vigor. The results showed that LDA model had the best effect, with an average accuracy of 92.9% for seed samples of different maturity and 97.8% for seed samples of different harvest years, and the average sensitivity, specificity and precision of LDA model could reach more than 90%. The average accuracy of nCDA in identifying dead seeds with no vigor reached 93.3%. In identifying the seeds with high vigor and predicting the germination percentage of alfalfa seeds, it could reach 95.7%. In summary, the use of Multispectral Imaging and multivariate analysis in this experiment can accurately evaluate and predict the seed vigor, seed viability and seed germination percentages of alfalfa, providing important technical methods and ideas for rapid non-destructive testing of seed quality.

## 1. Introduction

Seeds are the most fundamental means of agricultural production [1], and seed vigor is a comprehensive index to measure seed quality [2], which is a comprehensive evaluation of the activity intensity and characteristics of seeds during germination and emergence, including seed germination percentage, emergence percentage, seedling growth potential, plant resistance and yield potential, etc. [3]. Seed vigor first increased and then decreased during seed growth, development and storage. In the seed development stage, seed vigor rises with the increase in seed maturity and reached the peak at physiological maturity [4]. In the storage stage after full maturity, seed vigor decreases with the storage time due to natural aging [5]. Understanding the changing rules of seed vigor and evaluating it in different stages accurately and quickly is of great guiding significance for judging the maturity of seed growth and developmental period for safe storage of seed after maturity. However, it is very difficult to evaluate and quantify seed vigor accurately and comprehensively, because of the complex characteristics mentioned above. Therefore, the evaluation and determination of seed vigor has always been a difficult problem in the field of seed quality testing. At present, the International Seed Testing Association (ISTA) recognizes the standard seed vigor testing methods include electrical conductivity test (for chickpea (*Cicer arietinum*), soybean (*Glycine max*), kidney bean (*Phaseolus vulgaris*), and pea (*Pisum sativum*), radish (*Raphanus sativus*), etc.), artificial accelerated aging method (for soybean seeds, etc.), controlled deterioration method (for Brassica seeds), radicle emergence (for corn (*Zea mays*), Oilseed rape (*Brassica napus*), radish seeds, etc.), and tetrazole staining (for soybean seeds) [6]. Except for the species mentioned above, most of the seed vigor evaluation is still in the stage of experimental research, and there is no definite standard testing method.

With the development of science and technology and interdisciplinary application, multispectral imaging (MSI), a new non-destructive physical technique, provides a new idea for seed quality evaluation. The technology can obtain spectral information at 19 different wavelengths (365, 405, 430, 450, 470, 490, 515, 540, 570, 590, 630, 645, 660, 690, 780, 850, 880, 940, 970 nm), and can integrate traditional vision and spectral technology, simultaneously obtain the spatial and spectral information of the object, quickly and accurately determine the surface characteristics of the target object and detect its internal chemical composition, and also display their differences and changes. In recent years, many scholars have explored the application of this technique in seed vigor determination. Multispectral imaging combined with LDA model can accurately distinguish aged and unaged cowpea (*Vigna unguiculata*) seeds, and it has a high overall correct classification rate in predicting high vigor seeds, medium vigor seeds and dead seeds [7]. Cong [8] combined the method of seedling evaluation with the nCDA model in the multispectral software, and then predicted the perennial ryegrass (*Lolium perenne*) seeds with high, medium and low vigor with high accuracy. Galletti et al. [9] used two-component principal component analysis (PCA) method for exploratory analysis of multispectral imaging data, selected the most significant 5 wavelengths among 19 wavelengths, and carried out high-precision classification of high and low vigor seed in QDA-based model. The accuracy was 86~95% in tomato seeds (*Solanum lycopersicum*) and 88~97% in carrot seeds (*Daucus Carota*). Liu et al. [10] used PCA, LS-SVM, BPNN and RF models to evaluate the seed quality of watermelon (*Citrullus lanatus*), and the results showed that there were significant differences between high-quality watermelon seeds and the other watermelon seeds (including dead seeds and low vigor seeds). Among the used models in this study, the classification accuracy of LS-SVM and RF model was high. These studies indicate that multispectral imaging has a good application prospect in seed vigor testing.

Alfalfa (*Medicago sativa*) is a perennial herb of the genus Alfalfa in the leguminous family, known as the “King of forage” due to its rich protein content [11]. Because of its good forage value, ecological value and medicinal value, it is widely cultivated in the world. Because of this, timely and accurately monitoring of its seed vigor is very important for planting, transportation and storage. “Artificial Accelerated Aging Determination of Vigor of Grass Seed Testing Procedures” (NY/T 3187-2018) is currently the only standard issued in China that can testing the seed vigor for eight grass species including alfalfa. However, it has the disadvantages of small scope of application, cumbersome operation steps and destructive to seeds. Therefore, based on the above research status and basis of seed vigor determination, this study explored and evaluated the ability and feasibility of alfalfa seed identification and testing based on multispectral imaging at different vigor levels, laying a foundation for exploring the application of multispectral imaging in seed vigor and other important quality items testing.

## 2. Materials and Methods

### 2.1. Materials

“Zhongmu No.1” alfalfa seeds from different harvest years (2004, 2008 and 2019) and different maturity stages (green ripe, yellow ripening and full ripe) were used as experimental materials. Seeds from different harvest years or maturity levels come from the Alfalfa seed Production Base (39°12′ N, 106°95′ E) of Institute of Animal Science, Chinese Academy of Agricultural Sciences, Saiusu, Etoke Banner, Inner Mongolia, at an altitude of 1150 m, with an average annual temperature of 8.5 °C, average annual relative humidity of 52%, and average annual precipitation of 225 mm. We found that alfalfa pods were formed from bottom to top, and the pods at Section 2, Section 3 and Section 4 below the top of alfalfa were relatively full at green ripe stage. Therefore, seed samples of the three stages were taken from the relatively full pods mentioned above, and dried indoors, the abnormal pods in the inflorescence were removed. The sample of green ripe stage was collected on 3 July 2021, yellow ripening stage was on 12 July 2021, and full ripe stage was on 31 July 2021. After harvest, the seeds were stored in forage seed Laboratory of China Agricultural University with an average temperature of 25 °C and 35% average humidity.

### 2.2. Multispectral Imaging

The multispectral imaging instrument VideometerLab4 (Videometer A/S, Herlev, Denmark) was used to take seed sample images. Calibration and light setup of the instrument should be carried out before taking seed samples. Absolute reflectance was calibrated with a bright and dark reference object and geometrically aligned with a Dotted plate. The 660/700 nm and 405/600 nm excitation/emission combinations were then added to the default light settings to capture fluorescent images of chlorophyll A and chlorophyll B [9,12]. This was a completely independent analysis, and the data obtained were not involved in multivariate analysis. The seed sample was placed on the bottom of the instrument sphere and a high-resolution, multispectral original image of 2056 × 2056 pixels was obtained in seconds (Appendix A). The raw image was obtained with 19 different wavelengths (365, 405, 430, 450, 470, 490, 515, 540, 570, 590, 630, 645, 660, 690, 780, 850, 880, 940, 970 nm), and then the original image was manipulated. The Blob tool of VideometerLab software was opened to separate the seed sample from the irrelevant background, and the area, aspect ratio, color index, saturation, mean spectral reflectance, pixel distribution histogram and other measurement indexes of each seed were obtained (Appendix A).

In total, 400 seeds were randomly taken from each batch, 50 seeds were placed in a Petri dish each time and then put into the spectrum instrument. The shape and spectral relevant data of every seed was extracted and exported. Seed germination test was conducted according to the order of seed placement on Petri dishes since the results of seed germination test and the multispectral data of every seed should be one-to-one correspondence. The test was conducted in September 2021.

### 2.3. Determination of Seed Germination Characteristics

Germination of alfalfa seeds harvested in different years (2004, 2008 and 2019) and harvested in 2021 at different maturity stages (green ripe, yellow ripening and full ripe) were conducted according to ISTA *International Rules for Seed Testing 2021* in 20 °C light 8 h and dark 16 h from October to November 2021. Data of 400 seeds taken at random from each sample were collected using multispectral imaging before germination test. Normal seedlings, abnormal seedlings, hard seeds, fresh ungerminated seeds and dead seeds were recorded in the last count, and the germination potential and germination percentage were calculated according to the following formula:Germination potential (%) = (number of normal seedlings at the first count/total number of seeds tested) × 100%;
Germination percentage (%) = (number of normal seedlings at the end count/total number of seeds tested) × 100%;

### 2.4. Artificial Accelerated Aging Determination of Seed Vigor

The germination percentage of accelerated aging seeds was determined according to *Artificial Accelerated Aging Determination of Grass Seed Vigor Testing Procedures (NY/T 3187-2018)*. Cover the upper layer of aging box with clean gauze of appropriate size to prevent small seeds from leaking out of the hole, spread the seeds evenly and flatly on the gauze, and put them into aging box. At least 400 seeds were placed in an aging box and held for 48 h at 42 °C and 90–100% relative humidity. After aging, the seeds were taken out immediately, and seed germination test was conducted within 1 h. The results of vigor determination were expressed as the percentage (%) of germinated normal seedlings in the total number of seeds tested per treatment at the end of germination.

### 2.5. Data Analysis

The spectral information and seed morphology information collected by MSI were used for multivariate analysis as follows:

Principal component analysis (PCA), as an exploratory technique of multivariate data analysis, identifies morphological characteristics of extracted seeds and hidden patterns in spectral data, and is used to obtain an overview of systematic changes in the data. It is used to explore the possibility of grouping seeds with similar morphology and spectrum. By selecting less component information from linear transformation of multiple variables, the goal of dimensionality reduction can be achieved with sufficient information. In this study, the linear kernel was used in SVM for classification.

Support vector machine (SVM) can better solve the problems of small samples and high dimensions by mapping data to high-dimensional space and constructing the optimal classification hyperplane for data classification. The learning strategy of SVM is to minimize the intra-class observation distance and maximize the inter-class observation distance.

Linear discriminant analysis (LDA) projects data from a high-dimensional space to a lower-dimensional space and ensures that the in-class variance of each category is small and the mean difference between classes is large. This means that high-dimensional data of the same category are projected into low-dimensional space and the same categories are grouped together, but different categories are far apart.

Random forest (RF) is a combinatorial classification model based on decision tree predictor, which has excellent accuracy and can process input samples with high dimensional features without dimensionality reduction. In the process of the model development with RF using the combined spectral and morphological data, the number of classification trees desired (*n*_tree_) was defined as 500, and the number of variables (*m*_try_) used in each tree to make the tree grow was 3.

The same number of seeds randomly taken from each seed lot was divided into the training set (70% of the seed sample) and the validation set (30% of the seed sample), respectively. The classification model was established by using the training set, and the validation set was used to verify the obtained model. Accuracy, sensitivity, specificity and precision were obtained by using the confusion matrix obtained from the validation set to evaluate the classification performance of SVM, LDA and RF models. The formulas are as follows:Accuracy = (TX + TY)/(TX + TY + FX + FY) × 100%;
Sensitivity = TX/(TX + FY) × 100%;
Specificity = TY/(TY + FX) × 100%;
Precision = TX/(TX + FX) × 100%.

In the formulas, X and Y represent two samples, TX represents the sample X with correct prediction, TY represents the sample Y with correct prediction; FX represents the sample Y that is predicted to be sample X; FY means sample X that is predicted to be sample Y.

Normalized canonical discriminant analysis (nCDA) is a transformation construction method, and the calculation behind it is more similar to PCA, but more supervised. Set the two or more categories to be distinguished on different layers to generate a single-band visual analysis image that will be a linear discriminator between two categories, one mainly positive and the other mainly negative.

*FactoMineR, E1071, MASS* and random Forest packages in R language were used for PCA, SVM, LDA and RF analysis and prediction. nCDA analysis was achieved by using MSI-Transformation Builder in Videometer Software V.3.22.0 (Videometer A/S, Herlev, Denmark).

## 3. Results

### 3.1. Germination Test of Alfalfa Seeds at Different Maturity and Harvest Years

The results of standard germination test showed that, the seed germination potential and germination percentage increased significantly with the increase in maturity. The germination percentage of the green ripe seed sample was only 11%, for the yellow ripening seed sample it was 24%, and to the full ripe stage, the seed germination percentage could rise to 55%, and the seed germination potential rose from 5% to 30%. On the other hand, with the increase in storage years after seed harvest, the seed germination potential and germination percentage decreased significantly. The germination percentage of seed sample harvested in 2019 could reach 93% and is clearly decreased (67%) in 2008, while the germination percentage of seeds harvested in 2004 was only 6%. Generally, the germination potential reduced during 10 years from 91% to 5% (Figure 1).

According to the results of the artificial accelerated ageing test of seed vigor, the seed germination percentage of three seed lots with different maturity showed a gradient increase, among which the germination percentage of green ripe seeds was the lowest, and at full ripe stage it was the highest. According to the test rules above, the seeds of full ripe stage were evaluated as high vigor seeds, those of yellow ripening stage were evaluated as medium vigor seeds, and those of green ripe stage were evaluated as low vigor seeds. Similarly, for seeds harvested in three different years, seeds harvested in 2004 were low vigor, seeds harvested in 2008 were medium vigor, and seeds harvested in 2019 were high vigor (Figure 2).

### 3.2. Chlorophyll Fluorescence Determination

The fluorescence intensity of chlorophyll A and B quantified by the fluorescence images of chlorophyll A and chlorophyll B captured by MSI shows that, among seeds of different maturity, the chlorophyll A value at green ripe stage is higher than that at yellow ripening stage and full ripe stage, and the difference of chlorophyll B value is not significant. The value of chlorophyll A in seeds of different harvest years basically remained at the same level with the increase in storage years, and the value of chlorophyll B decreased significantly with the increase in storage years in a gradient pattern (Figure 3).

### 3.3. Analysis of Morphological Data and Spectral Data

Fourteen morphological characteristics of each lot of seeds of different maturity and different harvest years were extracted by MSI and compared separately. The results showed that there were significant differences in 12 indexes among different maturity. The area, length, width and other indexes of green ripe seeds were significantly lower than those of yellow ripening seeds and full ripe seeds, and there were significant differences in shape parameters and color parameters among the three seed samples with different maturity. Saturation and hue of green ripe seeds was significantly lower than that of yellow ripening and full ripe seeds. (Table 1). There were a total of 13 indexes with significant differences among seeds from different harvest years, and CIELab L*, CIELab A* and CIELab B*, the three parameters related to color, were all significantly different among seeds from three different harvest years (Table 2).

In addition, by collecting and analyzing multispectral data of seed samples, it was also found that the average reflectance of alfalfa seeds with different maturity and harvest years varied greatly at the wavelength of 470~690 nm, and the mean reflectance of seeds with different maturity at the wavelength of 940–970 nm also showed great difference besides the wavelength range above (Figure 4).

### 3.4. Multivariate Analysis

The results of principal component analysis based on morphological and spectral characteristics showed that for seeds of different maturity, the first two principal components accounted for only 54.6% of the original variance, and the 95% confidence ellipses in the PC1/PC2 two-dimensional diagram of the three seed samples overlapped a lot. For seeds of different harvest years, the first two principal components account for only 52.3% of the original variance. Therefore, PCA could neither distinguish the three alfalfa seed samples with different maturity, nor the samples with different harvest years very well (Figure 5). However, LDA model had obvious discrimination effect. No matter for seeds of different maturity or harvest years, most of the seeds of the three vigor levels were separated with a large degree of separation, among which seeds of green ripe stage and seeds harvested in 2019 showed the most obvious differentiation from other seeds (Figure 6).

The LDA model had the best prediction effect among the three models, and it could distinguish three seed samples of different maturity with an accuracy of 96.3% (green ripe stage vs. yellow ripening stage), 85.8% (yellow ripening stage vs. full ripe stage) and 96.7% (green stage vs. full ripe stage), respectively (Table 3), with an average accuracy of 92.9%. The seeds of different harvest years could also be well distinguished with an accuracy of 97.1% (2004 vs. 2008), 96.7% (2008 vs. 2019) and 99.6% (2004 vs. 2019), respectively. The average accuracy reached 97.8%, while the sensitivity, specificity and precision were high, with an average of more than 97% (Table 4).

### 3.5. Prediction of Seed Emergence

In addition, we predicted the germination results of each seed according to the nCDA model in VideometerLab 4 software, and we verified and evaluated the accuracy of the prediction by using the actual results after the standard seed germination test. The standard germination results of seed samples with different maturity and harvest years were classified and recorded according to the categories of dead seed (D), fresh seed (F), abnormal seedling (A), hard seed (H) and normal seedling (N), and the prediction accuracy was calculated, respectively. Taking the prediction accuracy of seed potential germination percentage (N + H) as an example, the seeds in the multispectral image after background removal were divided into two layers: one layer was hard seeds and normal seedlings, and the other layer was other seeds/seedlings. After nCDA model analysis, the seeds marked red were better matched with normal seedlings and hard seeds. Seeds marked blue were predicted to be seeds/seedlings other than hard seeds or normal seedlings, and then compared with actual germination to calculate accuracy (Figure 7).

The nCDA prediction results showed that the prediction accuracy of dead seeds without vigor could reach 95.6% for seed samples with different maturity, while the prediction accuracy of hard seeds with high vigor and normal seedlings was only 68.8% and 77.5%, respectively. However, the prediction of accuracy of the sum of hard seeds and normal seedlings, that is, the seed potential germination percentage (N + H), could reach 96.8%. For seed samples of different harvest years, the prediction accuracy of dead seeds, hard seeds and normal seedlings was above 90.0%, which were 90.9%, 93.5% and 90.0%, respectively. The prediction accuracy of the sum of hard seeds and normal seedlings was slightly higher than that of separate prediction, reaching 94.6% (Table 5).

## 4. Discussion

In this experiment, samples were taken during different stages of seed development and storage in order to obtain seeds with different vigor levels. The results of germination test and artificial accelerated aging test were consistent with the expected results, indicating that the seed vigor of the samples at different development stages and different storage stages presented high, medium and low vigor gradient level.

Compared with other nondestructive testing techniques such as soft X-ray and near infrared spectroscopy, MSI can obtain spectral and spatial information of seeds at the same time, which makes the information more diversified. The fusion method of spectral data and morphological data and the screening of fusion data have always been a difficulty and research focus in the field of nondestructive testing technology [13]. The results of chlorophyll fluorescence imaging showed that the degradation of chlorophyll existed in both the development stage and storage stage of seeds. For alfalfa seeds, the fluorescence intensity of chlorophyll A was the highest at the green ripe stage. This stage has incomplete chlorophyll degradation and low physiological potential [14], the higher the fluorescence intensity, the lower the seed vigor [9]; Degradation of chlorophyll B molecules occurred at the storage stage after maturity, and the chlorophyll fluorescence intensity of high-vigor seeds was higher than that of low-vigor seeds, which was similar to the previous research results on peanut (*Arachis hypogaea*) [12].

In this experiment, key information with good discrimination was also screened out from morphological data and spectral data. The projected area, length and width of seeds increased with the increase in maturity, indicating a positive correlation between seed size and vigor at the seed development stage. This was consistent with the results of studies on the relationship between seed size and vigor in rice (*Oryza sativa*) [15] and sunflower (*Helianthus annuus*) seeds [16]. The area and vigor of seeds in different harvest years were negatively correlated, which was consistent with the research results on soybean [17] and corn [18]. CIELab is an international standard for color testing [19]. Among the three parameters obtained by multispectral imaging, L* represents brightness, A* represents the color change degree from green to red, and B* represents the color change degree from blue to yellow [20], which can be applied to judge the maturity and vigor of seeds [21]. For seeds of different maturity, L* increased with the increase in vigor, and A* and B* also had significant changes in three stages, but there was no regularity. For seeds of different harvest years, L* and B* decreased and A* increased with decreasing seed vigor. In general, CIELab L* is a reliable color index for evaluating the seed vigor of alfalfa. The mean spectral reflectance of seeds with different vigor levels varied greatly from 470 to 690 nm, which may be related to the change of seed color and chlorophyll content [22,23]. The mean spectral reflectance of seeds with different maturity levels varied from 940 to 970 nm, which may be related to the changes in lipid content and water content during seed development [24]. However, seeds in different harvest years were fully mature, so there was little difference at 940–970 nm.

Related studies using leguminous seeds as experimental materials showed that PCA combined with multispectral data was not effective in distinguishing six leguminous seeds [25], nor was it effective in distinguishing alfalfa seeds from sweet clover (*Melilotus officinalis*) seeds [26]. There was also no good effect in distinguishing alfalfa seeds of different cultivars or different natural aging degrees [27,28]. By contrast, LDA in the above experiments has a good effect, and the prediction ability of LDA is superior to SVM in distinguishing different leguminous seeds and different varieties of alfalfa. LDA was also superior to SVM and RF in distinguishing alfalfa seeds with different natural aging degrees, which is consistent with the research results of this study. These may be due to their different working modes. During the modeling process, SVM model mainly relies on near-infrared regional spectral information [26,27], LDA model mainly relies on the reflectance of 430~490 nm band in the visible spectral range to make a large contribution [26], while RF focuses on seed morphology and spectral indicators together [10].

In addition, in this study, the determination ability of multispectral imaging was extended to predict the alfalfa seed germination [28,29]. Seed vigor is uneven even among seeds of the same maturity or harvested in the same year, which makes it difficult to accurately determine this index. According to nCDA prediction results, the effect of distinguishing normal seedlings and hard seeds separately was not ideal, but the effect of distinguishing them by combining them into one group was very good. The statistical results of germination test also showed that many hard seeds were predicted to be normal seedlings, and many normal seedlings were predicted to be hard seeds, indicating that nCDA prediction model could not distinguish hard seeds and normal seedlings well. However, if hard seeds and normal seedlings were grouped together, they can be well distinguished from other ungerminated seeds and abnormal seedlings, which indicated that hard seeds and seeds developed into normal seedlings were classified into one category by nCDA model. Hard seeds are similar to normal seedlings and belong to high vigor seeds with strong germination potential [30]. Therefore, this method can be applied to predict high vigor seeds in leguminous seeds. In the actual work of the statistics of the germination percentage of leguminous seeds in the standard germination test, the sum of normal seedlings and hard seeds was usually reported as the final germination percentage results. Therefore, nCDA was also ideal for predicting alfalfa seed germination percentage. In addition, the experimental results showed that nCDA had a strong ability to predict dead seeds, with an average accuracy of 93.3%, which had great application potential in predicting seed viability.

For the testing of seed vigor, standard germination percentage and viability by using the traditional method specified in the current standard, not only the duration of the test is long, but also the accuracy of the test results depends on the experience of analysts to a large extent. For example, there are some errors in the determination of seed germination percentage, such as the differentiation between normal seedlings and abnormal seedlings, the determination of dyeing area and location. The use of MSI for seed germination percentage and viability test, can immediately obtain the results, greatly shortened the testing period, not only to complete the seed quality testing, but also to achieve the seed quality prediction. By combining computer image with spectral information and applying multivariate analysis, the error interference caused by subjective judgment difference was greatly avoided, and the judgment result was more objective and accurate. In addition, the whole test process is nondestructive, and it is of great application value especially for the quality monitoring of seed resource bank and the quality testing of expensive and rare precious seeds. Although multispectral imaging technology has a very good application prospect and many advantages in the field of seed quality testing, it also has some limitations and disadvantages. For example, the equipment is expensive, the application time is short, many species and testing items have not completed the mature modeling work, there is still a long way to go before standardized testing and application promotion. However, it is believed that as the cost of computer hardware and artificial intelligence decreases, the application value of multispectral imaging technology in seed quality testing will be immeasurable.

## 5. Conclusions

In conclusion, the morphological data and spectral information of seeds extracted by MSI, combined with the LDA model and nCDA model, can accurately measure the seed vigor, viability and germination percentage of alfalfa, so as to achieve the established goal of predicting the seed germination ability quickly and nondestructively. The above method is suitable for the determination of seed vigor level difference caused by different level of maturity during seed development, and also for the determination of differences in seed vigor level caused by natural aging during different storage years after physiological maturity, which has good applicability and representativeness. The chlorophyll fluorescence imaging technique also provides a new idea for early identification of seeds with high or low vigor. The results of this experiment provide an important reference and basis for further exploring the application of multispectral imaging technology in seed quality testing.

## Figures and Tables

**Figure 1 sensors-22-02760-f001:**
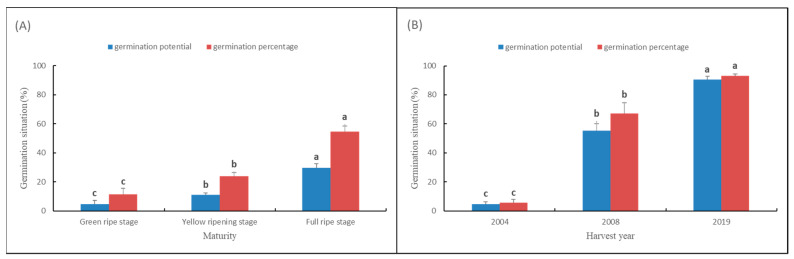
Germination potential and germination percentage of seeds at (**A**) different maturity levels and (**B**) harvest years. Note: Different lowercase letters indicate significant differences (*p* < 0.05).

**Figure 2 sensors-22-02760-f002:**
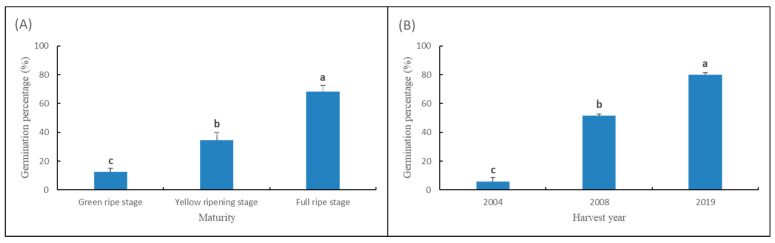
Artificial accelerated aging of seed vigor test at (**A**) different maturity levels and (**B**) harvest years. Note: Different lowercase letters indicate significant differences (*p* < 0.05).

**Figure 3 sensors-22-02760-f003:**
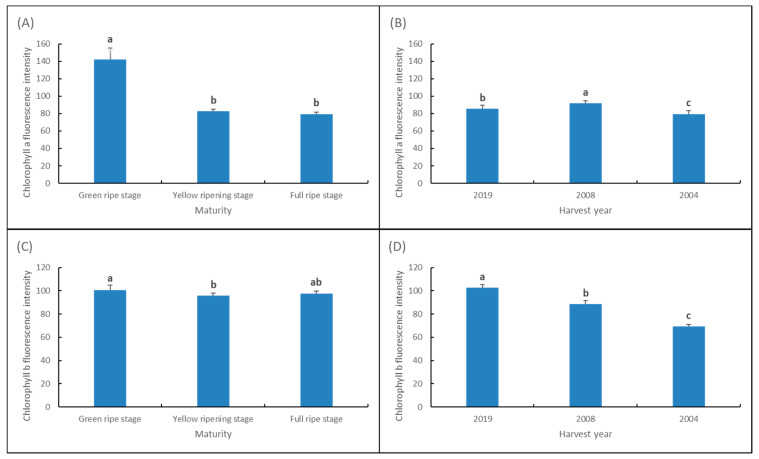
Chlorophyll A and chlorophyll B fluorescence intensity of seeds at different maturity levels and harvest years. (**A**) Chlorophyll A fluorescence intensity of seeds at different maturity levels. (**B**) Chlorophyll A fluorescence intensity of seeds at different harvest years. (**C**) Chlorophyll B fluorescence intensity of seeds at different maturity levels. (**D**) Chlorophyll B fluorescence intensity of seeds at different harvest years. Note: Different lowercase letters indicate significant differences (*p* < 0.05).

**Figure 4 sensors-22-02760-f004:**
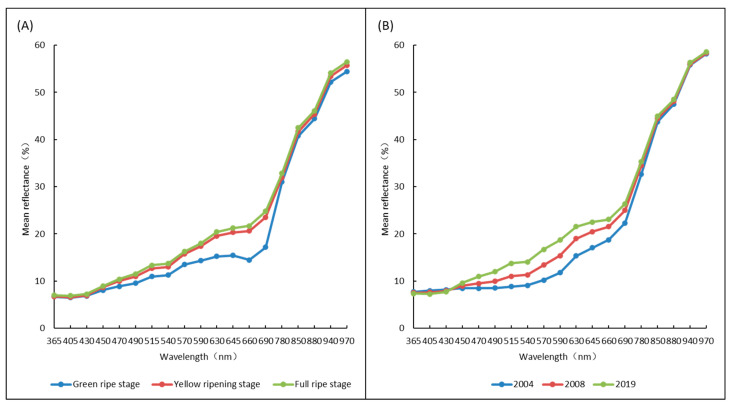
Mean spectral reflectance of alfalfa seeds at (**A**) different maturity levels and (**B**) harvest years.

**Figure 5 sensors-22-02760-f005:**
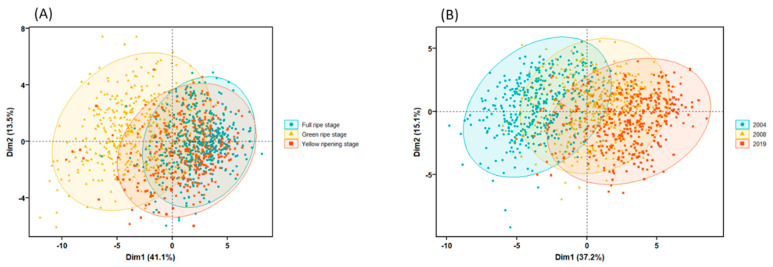
Principal component analysis based on multispectral data of alfalfa seeds of (**A**) different maturity levels and (**B**) harvest years.

**Figure 6 sensors-22-02760-f006:**
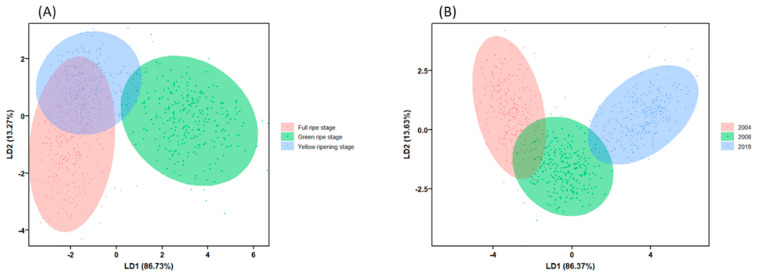
LDA model diagram based on morphological and spectral data of alfalfa seeds of (**A**) different maturity levels and (**B**) harvest years.

**Figure 7 sensors-22-02760-f007:**

nCDA image and actual germination of seeds. (**A**). nCDA images of seeds harvested at green ripe stage and actual germination at the last count of germination test. (**B**). nCDA images of harvested seeds at full ripening stage and actual germination at the last count of germination test.

**Table 1 sensors-22-02760-t001:** Comparative analysis of morphological characteristics of alfalfa seeds with different maturity levels.

Feature	Green Ripe Stage	Yellow Ripening Stage	Full Ripening Stage
Area (mm^2^)	2.09 ± 0.51 ^b^	2.75 ± 0.34 ^a^	2.77 ± 0.39 ^a^
Length (mm)	2.05 ± 0.29 ^b^	2.37 ± 0.22 ^a^	2.37 ± 0.21 ^a^
Width (mm)	1.40 ± 0.17 ^b^	1.58 ± 0.10 ^a^	1.58 ± 0.13 ^a^
RatioWidthLength	0.69 ± 0.08 ^a^	0.67 ± 0.08 ^b^	0.67 ± 0.07 ^b^
Compactness Circle	0.66 ± 0.08	0.65 ± 0.08	0.65 ± 0.07
Compactness Ellipse	0.98 ± 0.01	0.99 ± 0.01	0.99 ± 0.01
BetaShape_a	1.64 ± 0.17 ^a^	1.57 ± 0.15 ^b^	1.55 ± 0.13 ^c^
BetaShape_b	1.51 ± 0.17 ^a^	1.49 ± 0.13 ^b^	1.48 ± 0.13 ^c^
Vertical Skewness	−0.07 ± 0.05 ^c^	−0.05 ± 0.04 ^b^	−0.04 ± 0.03 ^a^
CIELab L*	39.27 ± 5.45 ^c^	46.05 ± 3.91 ^b^	47.47 ± 3.16 ^a^
CIELab A*	6.76 ± 3.77 ^c^	9.62 ± 2.12 ^a^	9.24 ± 1.43 ^b^
CIELab B*	29.45 ± 4.80 ^c^	33.35 ± 2.67 ^a^	33.10 ± 3.30 ^b^
Saturation	30.98 ± 4.66 ^b^	34.70 ± 2.41 ^a^	34.25 ± 3.19 ^a^
Hue	1.21 ± 0.62 ^b^	1.28 ± 0.15 ^a^	1.30 ± 0.05 ^a^

Note: Different lowercase letters in the same line indicate significant differences, while the same letters indicate no significant differences (*p* < 0.05).

**Table 2 sensors-22-02760-t002:** Comparative analysis of morphological characteristics of alfalfa seeds in different harvest years.

Feature	2019	2008	2004
Area (mm^2^)	2.53 ± 0.41 ^b^	2.88 ± 0.47 ^a^	2.94 ± 0.46 ^a^
Length (mm)	2.29 ± 0.23 ^b^	2.48 ± 0.26 ^a^	2.47 ± 0.24 ^a^
Width (mm)	1.48 ± 0.13 ^c^	1.54 ± 0.14 ^b^	1.59 ± 0.14 ^a^
RatioWidthLength	0.65 ± 0.07 ^a^	0.63 ± 0.07 ^b^	0.65 ± 0.07 ^a^
Compactness Circle	0.62 ± 0.07 ^a^	0.60 ± 0.07 ^b^	0.62 ± 0.07 ^a^
Compactness Ellipse	0.99 ± 0.01	0.98 ± 0.01	0.99 ± 0.01
BetaShape_a	1.50 ± 0.14 ^a^	1.46 ± 0.15 ^b^	1.50 ± 0.14 ^a^
BetaShape_b	1.43 ± 0.12 ^a^	1.39 ± 0.13 ^b^	1.42 ± 0.13 ^a^
Vertical Skewness	−0.04 ± 0.03	−0.04 ± 0.03	−0.04 ± 0.03
CIELab L*	48.33 ± 4.05 ^a^	43.72 ± 4.39 ^b^	35.75 ± 4.44 ^c^
CIELab A*	10.26 ± 2.56 ^c^	14.25 ± 2.74 ^b^	16.48 ± 2.01 ^a^
CIELab B*	29.71 ± 2.83 ^a^	27.54 ± 2.98 ^b^	19.70 ± 4.53 ^c^
Saturation	31.37 ± 2.42 ^a^	31.05 ± 2.42 ^a^	25.91 ± 4.29 ^b^
Hue	1.24 ± 0.09 ^a^	1.09 ± 0.10 ^b^	0.87 ± 0.09 ^c^

Note: Different lowercase letters in the same line indicate significant differences, while the same letters indicate no significant differences (*p* < 0.05).

**Table 3 sensors-22-02760-t003:** Prediction of alfalfa seeds with different maturity levels by LDA, SVM and RF models.

Model	Index	G vs. Y	Y vs. F	G vs. F
LDA	Sensitivity (%)	94.2	87.4	97.5
	Specificity (%)	98.3	84.3	95.9
	Precision (%)	98.3	84.6	95.9
	Accuracy (%)	96.3	85.8	96.7
SVM	Sensitivity (%)	95.0	89.1	95.8
	Specificity (%)	96.6	81.0	92.6
	Precision (%)	96.6	82.2	92.7
	Accuracy (%)	95.8	85.0	94.2
RF	Sensitivity (%)	91.7	82.4	99.2
	Specificity (%)	95.0	77.7	93.4
	Precision (%)	94.9	78.4	93.7
	Accuracy (%)	93.3	80.0	96.3

Note: G stands for green ripe seed, Y stands for yellow ripening seed and F stands for full ripe seed.

**Table 4 sensors-22-02760-t004:** Prediction of alfalfa seeds harvested in different years by LDA, SVM and RF models.

Model	Index	2004 vs. 2008	2008 vs. 2019	2004 vs. 2019
LDA	Sensitivity (%)	98.3	97.5	100.0
	Specificity (%)	95.9	95.9	99.2
	Precision (%)	95.9	95.9	99.2
	Accuracy (%)	97.1	96.7	99.6
SVM	Sensitivity (%)	94.1	97.5	99.2
	Specificity (%)	96.7	93.4	99.2
	Precision (%)	96.6	93.5	99.2
	Accuracy (%)	95.4	95.4	99.2
RF	Sensitivity (%)	95.0	87.4	97.5
	Specificity (%)	95.9	89.3	98.3
	Precision (%)	95.8	88.9	98.3
	Accuracy (%)	95.4	88.3	97.9

**Table 5 sensors-22-02760-t005:** Seed germination prediction of different maturity and harvest years based on nCDA.

Sample	Classification	Actually Number of CS	Correctly Predict Number of CS	Actually Number of RS	Correctly Predict Number of RS	Accuracy of Prediction(%)
Maturity	D	173	160	1027	987	95.6
	F	2	2	1198	1089	90.9
	A	17	14	1183	1033	87.3
	H	649	451	551	375	68.8
	N	359	232	841	698	77.5
	N + H	1008	982	192	179	96.8
Harvest year	D	406	378	794	713	90.9
	F	10	9	1190	1019	85.7
	A	26	15	1174	904	76.6
	H	80	57	1120	1065	93.5
	N	678	647	522	433	90.0
	N + H	758	719	442	416	94.6

Note: CS indicates corresponding classified samples, and RS for remaining classified samples.

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
