# Peer review of "Non-Destructive Testing of Alfalfa Seed Vigor Based on Multispectral Imaging Technology"

_sensors, 2022, doi:10.3390/s22072760_

Round 1
Reviewer 1 Report
The manuscript studies the possibility of prediction vigor of alfalfa seeds by means of Multi Spectral Imaging (MSI) and multivariate analysis. Both reflectance and fluorescence images in the Visible range are acquired. Images are processed and 14 morphological or colorimetric characters are extracted for each seed. Seed vigor is assessed by means of germination tests.
Two different studies are carried out: seeds from 2021 harvest at three different ripening stages, and stored seeds from three different harvests (2004, 2008, 2019). In the first part of the paper several classifiers are tested for discriminating different ripening stages or different harvests. In the second part Normalized canonical Discriminant Analysis (nCDA) was used for predicting seed vigor.
The topic is interesting because seed vigor is a comprehensive index of seed quality. The possibility of assessing it quickly and non-destructively, without lengthy germination tests, would be very useful for seed traders.
The study is adequately motivated. Presentation of analysis is generally clear, but with some unclear point that need fixation (see below). Graphic material and reference list are satisfactory.
Here is a list of unclear points in order of occurrence:
The expression at line 104: “then dried the samples indoor” is unclear, because it seems referring to removed pods, which would be strange. I suggest rephrasing the whole sentence. For example: “samples from the three stages were taken from the relatively full pods mentioned above, and dried indoor. Abnormal pods in the inflorescence were removed”.
A brief explanation of what the “Blob tool” is should be given. It looks to be an instrument of “Adobe Illustrator” (and maybe other software), but I had to search on Google for discovering it. It would also be advisable to say what software authors used for manipulating images.
Some more details about classified algorithms could be interesting for the reader. For example, what kind of kernel was used in SVD? Have different kernels been tested for achieving better performance?
At line 176 authors say that dataset was divided in training and validation sets. However, in Tables 3 and 4, a single set of statistics is given for each classifier. Are they training or validation statistics? It should be explicitly said in the paper. It would also be beneficial for the paper to show both statistics, allowing an assessment of robustness.
At lines 265-266 authors say that analysis was based on “morphological and spectral characteristics”. If the word "spectral" refers to CIELab coordinates I would suggest changing it with “colorimetric”. Otherwise, it seems that full reflectance spectra were included in the predictor matrix. On the other hand, is spectra are effectively included in the matrix authors should explain how they manage the pre-processing issue. Morphological data are heterogeneous and require auto-scaling. However auto-scaling is not recommendable in spectroscopic analysis, because it amplifies weak and noisy bands.
Another unclear point regard fluorescence measurements. Are fluorescence data included in the predictor matrix or is this a completely separated analysis? It should be explicitly stated in the text.
At lines 307-309 authors say than nCDA marks normal seedling/ hard seed with blue and the other with red. However, in figure 7, it looks that blue points correspond to locations where no seedlings are visible in the corresponding photo. Is it possible that “blue” and “red” has been exchanged in the text?
For the above mentioned reasons, I recommend a minor revision.
Reviewer 2 Report
The paper presents a method for testing seed vigor using multispectral imaging technology and several mathematic tools (PCA, LDA, SVM, RF and nCDA).
The authors should present some figures of the multispectral images used.
The accuracy evaluation method is not clear. Did the authors used Confusion Matrices on control samples? This aspect must be clarified.
Reviewer 3 Report
This paper aimed to investigate the possible use of different learning algorithms such as principal component analysis (PCA), linear discriminant analysis (LDA), support vector machine (SVM), random forest (RF), and normalized canonical discriminant analysis (nCDA) in predicting the alfalfa seed vigor. Based on the results, LDA had the highest prediction ability over other tested algorithms. The result of this study seems interesting for a broad range of readers and may have the potential to publish. However, there are some points needed to be addressed before publishing in this journal:
Multispectral bands were captured in this study, but you did not mention the captured bands in the abstract. It is unclear whether you captured blue, green, red, red-edge, and NIR or only RGB.
I am not convinced that LDA had the better performance based on its accuracy itself. There is a huge chance of a false-positive rate, and you need to provide more evaluation parameters like Precision and Recall.
Line 25, “Multispectral Imaging combined with multivariate analysis” algorithms cannot be combined with inputs. Algorithms can analyze inputs. Maybe you mean “the use of Multispectral Imaging and multivariate analysis”.
Some of the keywords already exist in the title. Please add different keywords than those who are already in the title.
Line 56-58, do you mean hyperspectral reflectance? You mentioned a range, but multispectral has specific bands.
Line 61-78, you need to explain machine learning algorithms in more detail here. Provide some examples of their use in agriculture. Why did you decide to use those algorithms? What was the gap? Then you can move to the objectives. There are good references for that, such as https://doi.org/10.3390/molecules26072053.
It is clear that the output is numeric or discrete. If you had different classes, you could use SVM. Otherwise, you should use SVR(Regression).
Overall, algorithms need more explanation. You need to provide more detail in each tunning parameter in each algorithm.
Why are you not using recall, precision, F-measure, and MCC? Please add this information if you have.
How many folds did you use for cross-validation? Did you split your dataset into training and validation datasets?
What software did you use? I got confused in M&M.
The discussion part needs to be improved significantly. You need to explain more about each tested algorithm and compared the results with previous studies. Also, bring the strengths and the limitations of this research into the discussion and discuss them.
Please share the codes and datasets that everyone can repeat the experiment.
Reviewer 4 Report
The article refers to modern technology coupled with appropriate statistics to identify seed viability and vigor. It is well designed and performed although the writing needs some improvement.
The issue is rather not break through but is useful as an application which is applied for several years. The same authors have published in the same journal two more article on the same subject. Again, the level of innovation is rather low but the application is useful and is worthy of publication.
Below you can find some point which may need a second look.
Line 11 …important index to evaluate seed quality. Replace with: to evaluate seed quality in plant species.
Line 12 … has always been a hot and difficult problem…. Replace with: a serious problem in the field
Line 12 … and rapid, and has a good application prospect... Replace with:... and rapid application having advantageous prospects in .......
Line 16 … and harvest years representing different... Replace with:... different level of maturity and different harvest periods (years), representing different vigor levels and age of seed, were.
.......
Line 21 … different harvest years, and… Replace with:... different harvest seasons…
Line 26 … seed germination percentage of alfalfa… Replace with:... percentages…
Line 36 … plant resistance and production potential, etc. [3].Seed vigor first in-creased and then decreased during seed growth, development and storage.… Replace with:... … plant resistance and yield potential, etc. [3]. Seed vigor first in-creased and then decreased during seed growth, development and storage.
Line 42 development period and safe storage… Replace with:...developmental period for safe storage
Line 52-53 Honestly, most of the seed vigor evaluation is still in the stage of experimental research, and there is no definite standard testing method.
Please rephrase since all referred methods are standard methods and have been used effectively for a long time as now.
Line 42 …other important quality items testing... Replace with:... seed quality traits.
Line 94-96 Zhongmu No.1" alfalfa seeds FROM different harvest years (2004, 2008 and 2019) 94 and different maturity STAGES (green ripe, yellow and full ripe) were 95 used as experimental materials. Seeds FROM different harvest years OR ma- 96 turity LEVEL come from the Alfalfa seed
Line 125-128 please rephrase… put into the spectrum instrument. The shape and spectral 125 relevant data of every seed was extracted and exported. Seed germination test was conducted according to (THE ORDER of SEED PLACEMENT ON PETRI DISHES) the seed putting order in the
petri dish on SINCE the results of seed germination test and the multispectral data of every seed should be one-to-one correspondence.
Line131 please specify: Germination of alfalfa seeds harvested in different years and harvested in 2021
Line 138 & 140 Please consider : (number of normal seedlings measured at the 1st measurement / Total number of seeds tested)
Line148…and aged for 48h at…HELD FOR
Line 150 please consider:...percentage (%) of germinated normal seedlings in the total number of seeds tested per treatment.
Lines 204-207: The germination percentage of seed sample harvested in 2019 could reach 93% AND IS CLEARLY DECREASED (67%) IN 2008 SEASON, while the germination percentage of seeds harvested in 2004 was only 6%. GENERALLY, THE GERMINATION POTENTIAL REDUCED DURING 1O YEARS FROM 91% TO 5% (Figure 1).
Figure 1. Germination potential and germination percentage of seeds at DIFFERENT MATURITY LEVELS AND HARVEST SEASONS. Note: Different lowercase letters indicate significant differences (P<0.05).
Lines 413-420 Please consider: The above method is suitable for the determination of seed vigor level difference caused by different LEVEL OF maturity during seed development, and also for the DETERMINATION OF DIFFERENCES IN SEED VIGOR LEVEL CAUSED BY natural aging during different storage years after physiological maturity, which has good applicability and representativeness. The chlorophyll fluorescence imaging technique also provides a new idea for EARLY identification OF seeds with high OR low vigor. THE RESULTS OF THIS EXPERIMENT provide important reference and basis for further exploring the application of multispectral
imaging technology in seed quality testing.

Author Response
Please find the answers in the attachment

Round 2
Reviewer 2 Report
The changes introduced improved the paper.
Reviewer 3 Report
Agree to publish
Reviewer 4 Report
The manuscript following editing is more clear and can be accepted for publication.